# iPTR: Learning a representation for Interactive program translation retrieval

## Abstract

Program translation contributes to many real world scenarios, such as porting codebases written in an obsolete or deprecated language to a modern one or re-implementing existing projects in one's preferred programming language. Existing data-driven approaches either require large amounts of training data or neglect significant characteristics of programs. In this paper, we present iPTR for interactive code translation retrieval from Big Code. iPTR uses a novel code representation technique that encodes structural characteristics of a program and a predictive transformation technique to transform the representation into the target programming language. The transformed representation is used for code retrieval from Big Code. With our succinct representation, the user can easily update and correct the returned results to improve the retrieval process. Our experiments show that iPTR outperforms supervised baselines in terms of program accuracy.

## 1 Introduction

Numerous programs are being developed and released online. To port codebases written in obsolete or deprecated languages to a modern one (Lachaux et al., 2020), or to further study, reproduce and apply them on various platforms, these programs require corresponding versions in different languages. In cases when developers do not make the translation efforts themselves, third-party users have to manually translate the software to their needed language, which is time consuming and error prone because they have to be the expert in both languages. Also, hard-wired cross-language compilers still require heavy human intervention for adaptation and are limited between some specified types of programming. In this paper, we discuss the potentials of data-driven methods that exploit existing big code resources to support code translation. The abundance of open source programs on the internet provides opportunities for new applications, such as workflow generation (Derakhshan et al., 2020), data preparation (Yan & He, 2020), and transformation retrieval (Yan & He, 2018). Code translation is another application that is gaining attention (Lachaux et al., 2020).

**Data-driven program translation.** Inspired by natural language translation, one line of approaches trains a translation model from large amounts of code data either in a supervised (Nguyen et al., 2013; 2015; Chen et al., 2018) or weakly-supervised fashion (Lachaux et al., 2020). Supervised approaches require a *parallel dataset* to train the translation model. In parallel datasets, programs in different languages are considered to be "semantically aligned". Obtaining the parallel datasets in programming languages is hard because the translations have to be handwritten most of the time. Besides massive human efforts, it is also a tricky problem to extract general textual features that apply to every programming language. A recent weakly-supervised method (Lachaux et al., 2020) pretrains the translation model on the task of denoising randomly corrupted programs and optimizes the model through back-translation. However, this method still relies on high-quality training data. Further, all these approaches directly reuse NLP approaches that neglect the special features of programming languages. Another potential approach is to use a retrieval system to obtain translation candidates directly from Big Code, which refers to well-maintained program repositories. However, existing code retrieval systems, such as Sourcerer (Linstead et al., 2009), lack the proper capabilities for code-to-code search and cross-language code retrieval. These methods ask users to give feedback on several preset metrics and questions (Wang et al., 2014; Dietrich et al., 2013; Martie et al., 2015; Sivaraman et al., 2019). None of these methods is tailored to cross-language retrieval.

In this paper, we propose an interactive program translation retrieval system IPTR based on a novel and generalizable code representation that retains important code properties. The representation not only encodes textual features but structural features that can be generalized across all imperative programming languages. We further propose a query transformation model based on autoencoders to transform the input program representation to a representation that has properties of the target language. Due to the succinct form of our code representation, IPTR can adapt the original query based on user annotations. This methodology can compete with existing statistical translation models that require a large amount of training data. In short, we make the **following main contributions**:

- We propose IPTR, an interactive cross-language code retrieval system with a program feature representation that additionally encodes code structure properties.
- We further propose a novel query transformation model that learns a refined code representation in the target language before using it for retrieval. This model can be trained in an unsupervised way but also improved through active learning.
- Based on our succinct code representation, we propose a user feedback mechanism that enables IPTR to successively improve its results.

## 2 SYSTEM OVERVIEW

We propose IPTR, an interactive cross language code retrieval system that supports program translation on multiple programming languages.

**Problem Definition.** Given a piece of source program $P_s$ written in language $L_s$, a selected target language $L_t$, and a large program repository $D_p = \{P_1, P_2, ..., P_n\}$, the goal is to find the best possible translation $P_t$ of $P_s$ in $L_t$ from $D_p$. The problem is to design an effective program feature representation that generalizes to many languages and can be updated through user feedback.

**Our solution: IPTR** The workflow of IPTR is shown in Figure 1. IPTR first constructs a succinct but informative feature representation for input programs (Section 3.1). Since the target is to identify a similar program in the target language, IPTR then applies a query transformation model (QTM) to transform this representation into an estimated feature representation of the translation (Section 3). The transformation model is trained in an unsupervised manner but can also be updated dynamically through active learning (Section 3.2.2). Finally, this new representation will be used as a query to retrieve the program that has similar features from the database. In addition, as an interactive system, IPTR allows the user to give feedback on the retrieved translation (Section 4). The user can either accept the result or make corrections. Based on our structured and informative feature representation, IPTR can easily and quickly adapt the query based on raw user corrections. Then with the new query, it may identify a more appropriate translation candidate in the second retrieval attempt.

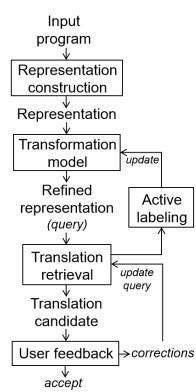

Figure 1: IPTR overview

## 3 PROGRAM REPRESENTATION

To retrieve a promising program translation from a large code database, IPTR needs an effective and efficient query. Directly retrieving based on raw code is impractical. In contrast to existing methods that generate queries based on either keywords (Linstead et al., 2009) or preset metrics and questions (Martie et al., 2015), IPTR generates a feature representation that effectively combines structural properties of the program and textual features. It further uses a query transformation model (QTM) to generate features in the target language.

### 3.1 BASIC ENCODING OF PROGRAM STRUCTURE AND TEXT

Due to the special and non-trival structure of programming languages compared to natural languages, we take both structural and textual features into consideration. The structural features of a program can be represented by its syntax tree where each tree node denotes a code construct.

Figure 2: Program representation

One can also use control flow graph (CFG) that captures the dependence between code basic blocks and procedures to represent the code behavior. However, our goal is to support program translation for any granularity of program, code behavior is hard to measure when the code fragment is not a complete code block. Further CFG is much more difficult to construct than syntax trees. As syntactical similarity also plays a significant role, we pick syntax trees as the basis of our representation to also capture the low-level syntactic structure within code blocks. One can also combine CFG and syntax trees to reserve more information. However this would trade-off the simplicity of the retrieval query. We show that with analysis of the static AST features we already achieve high program accuracy. The syntax tree can be either a concrete syntax tree (CST) or an abstract syntax tree (AST) (Alon et al., 2018; 2019; Chen et al., 2018). A CST depicts nodes with complete structural information, such as all the tokens in the code while the AST is more abstract and only displays structural or content-related details. For more details on CSTs and ASTs, we provide examples in the Appendix A.2.

As the CST is quite verbose and the AST does not generalize to multiple languages, we fall back on the low-level CST as a basis, and take the philosophy of AST as an inspiration to construct a unified abstract representation. Specifically, IPTR first simplifies the CST by removing semantically repetitive nodes such as equalityExpression and == and intermediate nodes and generates a more simplified but still informative syntax tree. As shown in Figure 2, a piece of Javascript code is first converted into a simplified syntax tree (details of simplifying CST are shown in Appendix A.2). Inspired by prior work (Alon et al., 2018), IPTR further simplifies it by extracting a set of one-dimensional paths that connect the program elements of the two-dimensional tree. Our method abstracts these paths by only keeping the three nodes that enclose the most critical information on a path: for each pair of leaf-nodes in the CST, IPTR keeps the nodes with their values and the root node of the statement. It drops all other intermediate nodes on this path. The extracted abstract paths of the JavaScript Program example are also shown in Figure 2. By simply matching nodes with similar names from different languages, IPTR can classify these paths into different types and generalize them to multiple programming languages. Thus, the structural feature of a program can be succinctly represented by the different types of paths $p_1, p_2, ..., p_j$ it contains.

To additionally incorporate textual features, IPTR treats extracted paths as plain text and keeps all the text tokens $t_1, t_2, ..., t_k$ appearing in each extracted path to generate text features. Finally, we can use these tokens (textual features) together with different types of paths (structural features) illustrated above as feature elements of programs. For each input program, we generate a feature vector consisting of the feature element frequencies in that program. Let $f$ be the occurrence frequency of feature elements, then the final feature representation will be $[f_{e_1}, f_{e_2}, ..., f_{e_n}] = [f_{p_1}, f_{p_2}, ..., f_{p_{n_p}}, f_{t_1}, f_{t_2}, ..., f_{t_{n_t}}]$. In our experiment, the number of different path types is about 5,000 on average for each language pair. The number of tokens can be restricted by a hyper-parameter max_vocab to trade off effectiveness and efficiency. Our default setting is 10,000. These elements can also be used as index keys to filter the database, and by calculating the similarity of feature vectors IPTR can retrieve the most similar program in target language, which is the translation candidate. Noted that we simplify the feature representation presented here due to space limitation. In actual IPTR, we also consider the dependencies between structural and textual features. In contrast to existing work (Cheng et al., 2016; 2017) that suggests to extract all the text from a program and treat it in isolation, IPTR considers textual features in strong dependency with the structural features to leverage more context from the structure. For the sake of simplification and to avoid computation overhead, IPTR only processes text that appears in the extracted paths. In this way, IPTR can run an efficient hierarchical retrieval mechanism: first calculating the structural similarity, then comparing the textual similarity in each common path type. We show the details of our comprehensive feature representation in Appendix A.3.

## 3.2 QUERY TRANSFORMATION MODEL

We can directly use the feature vector described in Section 3.1 as a query to retrieve translation candidate. However, the feature vector will still fail to accommodate some cross languages hurdles. For example, C# supports `goto` statement while its Java translation has to use `break` or `continue` with label instead of `goto`. In this case, IPTR cannot directly use the features of C# program to retrieve its Java translation. The result can be improved if the retrieval can be conducted based on the features of the translation. While translation of a complete program is our original problem and hard to solve, we propose a query transformation model (QTM) that solves a smaller problem. QTM transforms the original query (feature vector of the input program) into an optimized query (estimated feature vector of the translation). While the estimated feature vector is not a full program it is a better estimate for finding the correct program in the database.

### 3.2.1 MODEL DESCRIPTION

As shown on the left of Figure 3, in the online phase, to translate from language $L_s$ to language $L_t$, IPTR extracts the features as $F_s$ of a program from source language $L_s$ and feeds it into the QTM. In the QTM, a one layer encoder (red in Figure 3) maps the original feature vector to a low dimensional latent space and produces a shorter hidden vector $H$. Then $H$ is reconstructed to the estimated translation feature vector $F_t$ by a one layer decoder (yellow in Figure 3). $F_t$ will be used as query to retrieve potential translation in target language $L_t$.

Since there is no available training data for QTM, we leverage an unsupervised method - autoencoder (AE) to train the encoder and the decoder. An AE is an encoder-decoder that aims to reproduce its input. That is, it encodes the input to a hidden vector, then reconstructs the input from this hidden vector. In IPTR, we exploit this property to learn the weights of the encoders and decoders separately. As shown on the right of Figure 3, in the offline phase, for each programming language $L_i$ in the database IPTR trains a separate $AE_i$ on all programs that are written in this language. Thus it obtains a pair of $Encoder_i$ and $Decoder_i$ for each programming language. For the actual translation task, we combine the appropriate encoder and decoders depending on the source and target language of a translation task. We can see in the QTM in Figure 3, it selects the encoder $Encoder_s$ of the source language $L_s$ from $AE_s$, which is trained in offline phase, to transform $F_s$ into the hidden layer representation. And it picks the trained decoder $Decoder_t$ of the target language $L_t$ from $AE_t$ to estimate the $F_t$. This way, we can build a pretrained model with an encoder that learns significant information from the feature vector of the input program and an decoder that can generate features of its translation.

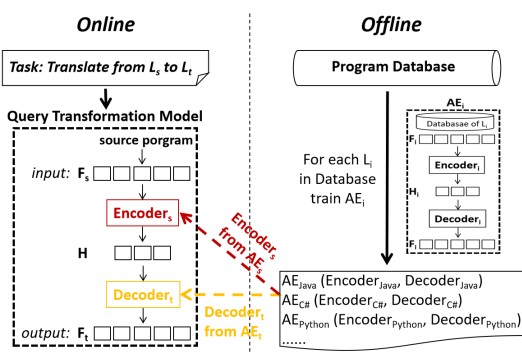

Figure 3: Query transformation model (QTM)

### 3.2.2 SAMPLING STRATEGIES FOR ACTIVE LEARNING

To make the QTM more suitable for specific real-world code data, we also design an active learning mechanism to enable the user to fine-tune the mixed encoder model. As shown in Figure 1, during each translation retrieval, the most useful programs are selected with a sampling strategy for user labeling. IPTR will ask the user to give the correct translation as its label. We propose an aggregation of four sampling strategies to approximately measure the informativeness of the given program as following (We take program $A$ that contains a set of feature elements $\{e_1, e_2, ...\}$ as an example):

1. **Coverage sampling.** It picks the programs that cover more different feature elements, which may reveal more information. We consider over half of the average feature elements amount as high coverage. In a database, if the average amount of feature elements contained in a program is $\lambda$ and program $A$ contains more than $\lambda/2$ different feature elements, it is a qualified sample.
2. **Rarity sampling.** Rarity sampling considers programs with rare feature elements, i.e. programs that contain features that appear in at most $\epsilon\%$ of the database programs. For example, if feature

element $e_1$ from program $A$ appears in $x\%$ ($x < \epsilon$) of the database programs, $A$ is a qualified sample.

3. **Uncertainty sampling.** Uncertainty sampling picks retrieved programs with low certainty, i.e., lower similarity score than 75%. For example, if program $B$ is the top retrieved translation of program $A$, but their similarity score is 50%, which is lower than 75%, program $A$ is a qualified sample.

4. **Random sampling.** It randomly selects a program for labeling (Zhu et al., 2007).

After running the above four sampling strategies for an input program, IPTR employs query-by-committee method to aggregate the results. With this approach, we make sure to have incorporated a diverse set of characteristics that might be relevant for sampling. The final decision is made by selecting program data where the largest disagreement occurs among those sampling strategies. The level of disagreement of a program $x$ can be measured by vote entropy $VE$ (Dagan & Engelson, 1995):

$$VE(x) = -\frac{V(x)}{N_s} \log \frac{V(x)}{N_s} - \frac{N_s - V(x)}{N_s} \log \frac{N_s - V(x)}{N_s} \tag{1}$$

$V(x)$ is the number of sampling strategies that select/vote $x$ as valuable sample. $N_s$ is the number of sampling strategies, which equals 5 in our case. Programs with higher vote entropy are returned as samples. After a task was successfully completed a new label can be stored for the QTM at hand.

## 4 QUERY ADAPTION BASED ON USER FEEDBACK

IPTR can also adapt a query based on user's feedback on returned results to improve the query. Existing interactive code retrieval methods ask the user to give feedback on the preset metrics and questions (Nie et al., 2016; Sivaraman et al., 2019). On contrary, IPTR directly leverages user's corrections to the retrieved results as feedback circumventing the efforts needed to pose the right questions. The simplest form of using the user corrections is to just use the feature vector of the corrected program. However, we also can make use of the fact that user corrections lead to manually curated features. To reflect this in our feature representation, we extend it with a weighting scheme. As shown in Figure 4, we obtain a new feature representation $R$, where each element consists of a feature element $f_{e_i}$ and its weight $w_i$. The initial weights are uniform. After the user makes corrections to the result, IPTR featurizes the correction the same way and compares it with the original code to generate the weights. We classify corrections into three categories and the weights are tuned accordingly:

- **Emphasize.** If the correction increases feature element $f_{e_i}$ that already exists in $R$, the weight of $f_{e_i}$ will be increased.
- **Add.** If the correction adds feature element $f_{e_i}$ that does not exist in $R$, $f_{e_i}$ and its initial weight $w_i$ will be added into $R$.
- **Delete.** If the correction decreases feature element $f_{e_i}$ in $R$, its weight $w_i$ will also be decreased.

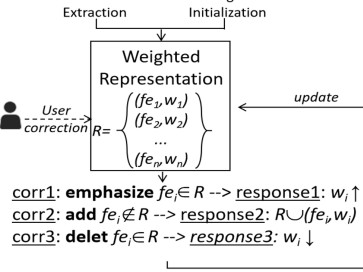

Then the feature representation is updated and used for a new round of retrieval. When calculating the similarity between query and instance in database, we consider an instance is better if it has higher similarity in features with high weights. We show an example of user feedback in Appendix A.4.

Figure 4: User feedback mechanism

## 5 EXPERIMENTS

In this section, we first compare different variations of IPTR with existing work from data-driven program translation and code search. We further evaluate IPTR's performance for different languages and discuss the user feedback. In addition, we also conduct an experiment on a large real-world dataset without ground truth to show the practicality of IPTR. This experiment is shown in Appendix A.5. All our experiments are conducted on a PC with an Intel Xeon E5-2650 v2 2.60GHz CPU and an NVIDIA Tesla K40m GPU.

**Dataset.** Datasets with ground truth are generally scarce. We run our experiments on the parallel dataset used in previous work with ground truth on Java to C# translations (Nguyen et al., 2013; 2015; Chen et al., 2018) and the dataset GeekforGeeks provided by Lachaux et al. (2020). The former was built based on several authoritative open source projects, such as Lucene, ANTLR, and JGit, which have both official Java and C# versions that have been correctly aligned to 39.797 matched methods. The latter gathered and aligned 698 coding problems and their solutions in several programming languages. We train the AEs of QTM on a database generated from the Public Git Archive (PGA) - a database with more than 260,000 top bookmarked Git repositories (web, c). We cleaned the dataset beforehand by gradually removing duplicates at file level, and files that cannot be successfully parsed due to format, errors, version compatibility or other issues. Finally, we obtain a dataset with the size of 260GB. We split all the files into methods or functions.

**Metric.** To measure effectiveness, we use *program accuracy* as proposed by prior work (Nguyen et al., 2015; Chen et al., 2018). Each time, we pick one program from the dataset and try to retrieve its translation from the database. *Program accuracy* is the percentage of the retrieved translations that are exactly the same as the ground truth in the dataset. Note that, it is an underestimation of the true accuracy based on semantic equivalence. That is, program and its translation should have the same functionality, but minor differences, such as variable names and writing habits, can be tolerated. Since we use a parallel dataset for evaluation, we can directly use the ground truth.

## 5.1 COMPARISON WITH BASELINES

We compare IPTR with four program translation baselines (**1pSMT** (Nguyen et al., 2013), **mppSMT** (Nguyen et al., 2015), **Tree2tree** (Chen et al., 2018), **TransCoder** (Lachaux et al., 2020)), which use a translation model to generate the results. The supervised baselines use 90% matched method pairs as training data to predict the translations for the rest of programs. We used the openly available implementation of **Tree2tree**. For TransCoder, we follow their method to pre-train the cross-language model on the Public Git Archive dataset (30GB of Java and C# data). For the program translation baselines 1pSMT, mmpSMT and Tree2tree, we report the results from their work on the same dataset as their code and configurations are not available. Finally, we report the results of two code search baselines **Sourcerer** (Linstead et al., 2009) and **CodeHow** (Lv et al., 2015)).

We generate different versions of IPTR with variations in interaction and feature representation:

- **Interative versions**: We discuss three different interactive versions of IPTR. All of which use the same feature representation and the QTM module. $\text{IPTR}_{\text{AL+FB}}$ is the full fledges interactive system with active learning for QTM and user feedback as described in Table 1. $\text{IPTR}_{\text{AL}}$ and $\text{IPTR}_{\text{FB}}$ only use the active learning component or the user feedback component, respectively.
- **Representions**: PTR refers to the basic program translation retrieval module with the feature vector of input program as query (Section 3.1). PTR+QTM uses the QTM module without active learning on top of PTR. We further analyze $\text{PTR}_{\text{WORD2VEC}}$ and $\text{PTR}_{\text{CODE2VEC}}$ as two feature representations variations of PTR based on **Code2vec** (Alon et al., 2019) and **Word2vec** (Mikolov et al., 2013a), respectively.

Table 1 shows the results and the degree of supervision. We observe that holistic IPTR with at most one user correction per task and optimized QTM outperforms all the baselines. The improvement in program accuracy is ranges from 19.5% to 65.5%. As expected, the code search baselines perform poorly because they are designed for retrieval with more accurate and detailed input than raw program code. The results of partial components of IPTR are also encouraging. PTR and PTR+QTM, which do not leverage any supervision, outperform the **Tree2tree**, which shows the effectiveness of our program feature representation and translation retrieval methodology. Leveraging the QTM successfully improves the result by 7.5% showing that generating features in the target language is more promising than using features of the source language. If the QTM is trained by active learning, the accuracy can increase by 8.5% ($\text{IPTR}_{\text{AL}}$). The table also shows that the Word2vec and Code2vec variants of PTR cannot outperform PTR with our proposed feature representation based on structural features. Word2vec is designed for natural languages so that it can not capture the special features of programming languages. Although Code2vec is designed specifically to represent code, their model can only be trained within a same programming language, which makes it less suitable for cross-language similarity comparisons.

Table 1: Comparison of different methods on program accuracy (PA) and supervision extent

| Genre | Method | Description | PA | Supervision Extent |
|---|---|---|---|---|
| Data-driven program translation | 1pSMT | Phrase-based SMT | 24.1% | fully supervised |
| | mmpSMT | multi-phase phrase-based SMT | 41.7% | |
| | Tree2tree | tree-to-tree neural networks | 70.1% | |
| | TransCoder | weakly-supervised neural translation | 49.9% | weakly supervised |
| Code search system | Sourcerer | Lucene-based code search, free-text queries | 13.5% | - |
| | CodeHow | free-text queries | 13.5% | |
| Variations of IPTR | PTR | | 71.1% | no labels (directly retrieve translation with input) |
| | PTR$_{\text{WORD2VEC}}$ | word2vec as queries | 67.7% | |
| | PTR$_{\text{CODE2VEC}}$ | code2vec as queries | 63.4% | |
| | PTR+QTM | | 78.6% | |
| | IPTR$_{\text{AL}}$ | | 87.1% | 80 labels for QTM |
| | IPTR$_{\text{FB}}$ | | 79.3% | At most 1 correction per task |
| | IPTR$_{\text{AL+FB}}$ | the full system | **89.6%** | combination of IPTR$_{\text{AL}}$ and IPTR$_{\text{FB}}$ |

Table 2: Comparing program accuracy with Transcoder on GeeksforGeeks dataset

| | C++-Java | C++-Python | Java-C++ | Java-Python | Python-C++ | Python-Java |
|---|---|---|---|---|---|---|
| Transcoder | 3.1% | 6.7% | 24.7% | 3.7% | 4.9% | 0.8% |
| PTR | 69.2% | 65.3% | 70.9% | 59.3% | 55.4% | 54.2% |
| PTR+QTM | 76.6% | 74.2% | 78.1% | 68.1% | 59.2% | 59.6% |
| IPTR$_{\text{AL}}$ | 87.2% | 79.5% | 84.8% | 72.5% | 66.2% | 68.8% |
| IPTR$_{\text{FB}}$ | 75.9% | 71.3% | 76.5% | 64.2% | 59.5% | 63.3% |
| IPTR$_{\text{AL+FB}}$ | 84.8% | 83.0% | 90.5% | 77.5% | 67.8% | 68.1% |

We further explored the supervision impact on IPTR. In this experiment, we let the user give at most one correction to each retrieved task. Without optimizing the QTM, IPTR$_{\text{FB}}$ slightly improves on PTR+QTM. The same improvement rate can be observed when comparing IPTR$_{\text{AL+FB}}$ to IPTR$_{\text{AL}}$, suggesting that AL and FB have independent influence on the results. Compared to the fully supervised methods 1pSMT, mmpSMT and Tree2tree, IPTR leverages very limited human supervision to achieve better results. In the first retrieval round, the user does not make correction to any wrong results, IPTR achieves 87.1% accuracy with only 80 labels for QTM. Also the reproduced weakly supervised approach TransCoder does not achieve better results than IPTR. Although some of the performance loss can be contributed to the fact that our training data is not as large as they report in their paper, we also observed that their model often generates invalid translations with regard to grammar. For example, it often mistakes the input type of a function for different languages. This phenomenon is also acknowledged in their own paper and can be attributed to the fact that only textual features have been used. IPTR avoids this problem by reusing existing code.

## 5.2 COMPARISON ON MULTIPLE PROGRAMMING LANGUAGES

We further compare IPTR and the state-of-the-art method Transcoder on the GeeksforGeeks benchmark provided by Lachaux et al. (2020). It contains groundtruth for Java, Python, and C++. Table 2 shows that the program accuracy of IPTR is significantly higher than Transcoder on their own datasets as also reported in their own paper. This is because of two reasons: (1) Our targets are different - Transcoder outputs machine-generated translations while IPTR directly retrieves existing programs as translation candidates, which makes IPTR always output syntactically correct programs. (2) Transcoder generally treats programming languages as plain text and aims to generate semantically similar programs. It does not encode the non-trival syntactical features of programming languages in their model. Thus, it has difficulties in generating results exactly the same as the ground truth. This experiment shows that IPTR is generalizable for multiple languages including dynamic languages, such as Python. Further it shows that it is easier to find translation candidates for grammatically similar programming languages, such as C++ and Java compared to Python as a high-level dynamic language.

## 5.3 INFLUENCE OF USER CORRECTIONS

In Table 1, we showed the influence of a single user correction on the result. We further investigate the number of user corrections required in order to retrieve the correct translations. We simulate users with the ground truth in our parallel dataset and for each returned result we fix the first different line between true result and returned result.

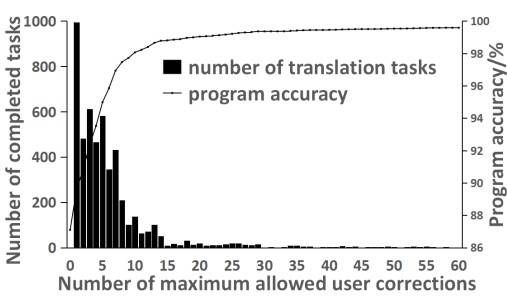

Figure 5: Required amount of user feedback

Figure 5 shows the number of required user corrections to obtain the correct result for all 5.134 translation tasks that failed in the first retrieval round and the improvement in the overall accuracy. We observe that in most cases, IPTR only requires a single user correction to successfully complete the translation task. About 98% of the failed retrieval tasks can succeed after 10 user corrections. Considering the average length of an input programs is 168 lines, we can conclude that with small amounts of user feedback the accuracy of IPTR can be significantly improved. Note that, we might even achieve better results if we do not restrict the users to fix the first difference each time. A real user might fix more significant errors that improve the success rate of the retrieval task. The average response time of IPTR to each user feedback is **9.4ms** using an efficient index structure.

## 6 RELATED WORK

**Data-Driven program translation.** Existing research mainly focus on building a translation model. Nguyen et al. (2013) directly applied the phrase-based statistical machine translation (SMT) model on the lexemes of source code to translate Java code to C#. In their follow-up work, they develop a multi-phase, phrase-based SMT method that infers and applies both structure and API mapping rules (Nguyen et al., 2015). But they are limited to languages that are similar on either structural or textual level, such as Java/C#. Chen et al. (2018) binarize the code tree using left-Child right-sibling and translate code by an LSTM-based encoder-decoder model. However, all these approaches require a large parallel dataset for training. In contrast to them, a recent work also proposes a weakly-supervised system TransCoder (Lachaux et al., 2020). They first train a cross language model for programming languages through predicting randomly masked words task, then acquire a pre-trained translation model from denoising randomly corrupted program task. Finally they improve this model through back translation. Although they do not need parallel translation data, this transfer learning method highly relies on the similarity of the data for pre-trained. And this approach is originally designed for natural language processing, which makes it produce results neglecting the features of programming languages. Our code retrieval system can support program translation without parallel dataset and violating the strict grammar rules of programming languages, which is a non-trivial problem in all the above methods.

**Interactive program retrieval.** In addition to the Sourcerer (Linstead et al., 2009) and StackOverflow search engines that were discussed in the introduction, there are some other work incorporating user interaction. Wang et al. (2014) refine query based on user's feedback on each result and re-order the rest ranking list. Nie et al. (2016) extract relevant feedback from StackOverflow for the initial query and reformulate it using Rocchio expansion. Dietrich et al. (2013) utilize a novel form of association rule mining to learn a set of query transformation rules from user feedback and use them to improve the efficacy of future queries. Martie et al. (2015) propose CodeExchange that uses the previous results to formulate the query so that user can find new results based on the previous characteristics. Sivaraman et al. (2019) propose an active learning system ALICE to iteratively refine a query based on positive or negative labels. All these approaches are designed for the mono-language setting and their queries are in natural language. IPTR performs cross language retrieval and directly uses raw code as input. Moreover, their method asks the user to feedback on the preset metrics and questions so that the result improvement largely depends on the subjective design from the developer. Due to the different understandings of different users, the feedback cannot be evaluated fairly and uniformly. IPTR directly uses user's corrections to the retrieved results as feedback and integrates an active learning based query transformation model to refine the query.

**Program representation.** By constructing program representations, one can enable the application of data processing to a wide range of programming-language tasks including program translation and code search. Kamiya et al. (2002) and Allamanis et al. (2016) treat a program as natural language and use the sequence of tokens as representation to detect code clones and summarize code. Allamanis et al. (2018) present a Gated Graph Neural Network in which program elements are represented by graph nodes and their semantic relations are edges in the graph to predict variable name and select correct variable. These methods rely on semantic knowledge, which requires more expert analysis and is not generalizable as the semantic analyses need to be implemented differently for every language. A recent approach uses paths in program's abstract syntax trees (AST) as code representation to predict program properties such as names or expression types (Alon et al., 2018). And they further leverage a tree-based neural network to encode these paths and generate more abstract representations (Alon et al., 2019). Yin et al. employ neural networks to express source code edits (Yin et al., 2019). However, these methods are not designed for translation retrieval. Some of them are too simple to effectively represent program's features (Kamiya et al., 2002; Allamanis et al., 2016). Others are too abstract for users to understand intuitively, which makes it impossible to easily interact with users (Allamanis et al., 2018; Alon et al., 2018; 2019; Yin et al., 2019). The program representation we use as query is more structured. It is not only informative for translation retrieval but also succinct for user interaction.

## 7 CONCLUSION

We presented ɪPTR, a novel interactive translation retrieval system that can support program translation. Different from traditional data-driven methods that train a translation model, ɪPTR directly searches for the most suitable translation candidate in existing code database. We propose a program representation based on a query transformation model that can learn a succinct but informative feature vector to retrieve the translation of an input raw program. ɪPTR can easily adapt the representation to user corrections for interactive retrieval improvements. Our experiments show that ɪPTR outperforms existing solutions in terms of effectiveness and requires no training data.

**Future work.** One important direction is to design a convenient user interface that allows user to make relevant corrections to translation suggestions and enable the system to converge faster to the desired result. Another possible direction to explore is other forms of user interaction that do not require the user to provide corrections in the target language.

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

## A    APPENDIX

### A.1    CST AND AST

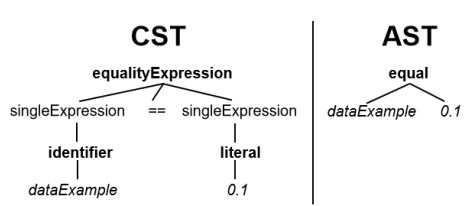

Figure 6: CST and AST of a simple JavaScript program

The structure of a program can be described by a syntax tree where each tree node denotes a construct occurring in the code, e.g., `if_stmt` in Python denotes an `if` construct. The tree can be either a comprehensive CST or a more abstract AST. CST is a tree representation of the grammar (rules of how the program should be written). It represents the source program exactly in parsed form. In other words, it defines the way programs look like to the programmer. There is an example in Figure 6. The CST of the simple JavaScript program is verbose with all the detail information about parsing the code. It keeps all the tokens in the program and their types, such as `literal`

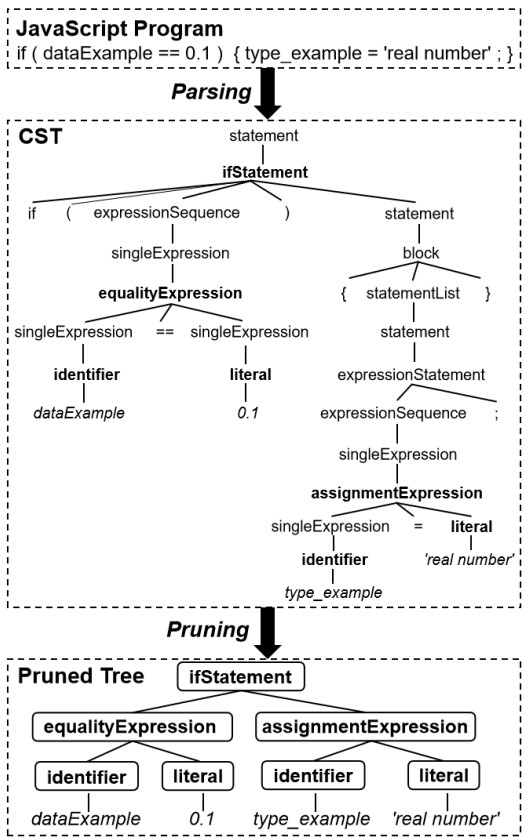

Figure 7: Generating simplified syntax tree

and `identifier`. And it reveals all the grammar rules, such as `equalityExpression: singleExpression == singleExpression`. On contrary, AST is a tree representation of the abstract syntactic structure of source code. Each node of the tree denotes a construct occurring in the source code. The syntax is "abstract" in the sense that it does not represent every detail appearing in the real syntax, but rather just the structural or content-related details. It defines the way the programs look like to the evaluator/compiler. As shown in Figure 6, the AST directly shows the `equal` structure without any detail information. It discards the intermediate structural information that would not contribute to semantics.

A.2    GENERATING SIMPLIFIED SYNTAX TREE.

IPTR first employs a left-to-right parser, such as ANTLR web (b), to parse the source code and generate the original CST. Figure 7 shows the CST of a raw JavaScript program. However, as shown in Figure 7, the original JavaScript code is simple while its CST is comparably intricate with all the details preserved. The large amount of nodes and branches will raise the computation complexity while having little contribution to the representation. For example, the node type `ifStatement` already specifically indicates the existence of an `IF` expression. Therefore its parent node type `statement` is not necessary as it does not reveal more information. On a syntax tree, all the nodes are already enclosed by their parent node. Thus, nodes of parentheses are not needed to reveal enclosure relationships. Further, node `==` is not necessary when node type `equalityExpression` already explains the content. Thus, IPTR follows two rules to prune the CST: (1) Node types that only have one child node are discarded, unless this child node represents specific content in the source code (Italics in Figure 7). For example, `statement` and `singleExpression` are discarded, while `identifier` and `literal` are retained. (2) If the parent node of a terminal node has more than one child node, this terminal node is removed. This is the case for symbols, such as `(` and `==`. The pruned tree in Figure 7 is much simpler than the original CST and retains all major structural features.

Table 3: The representation of the program from Figure 7

| Path type | Freq. | Text statistics |
|:---:|:---:|:---:|
| $P_1$ | 1 | {"data":1,"type":1,"example":2} |
| $P_2$ | 2 | {"data":1,"type":1,"example":2, "0.1":1,"real":1,"number":1} |
| $P_3$ | 1 | {"0.1":1,"real":1,"number":1} |
| $P_4$ | 1 | {"data":1,"example":1,"0.1":1} |
| $P_5$ | 1 | {"type":1,"example":1,"real":1,"number":1} |

### A.3 COMPREHENSIVE PROGRAM REPRESENTATION

In our comprehensive program representation, we not only consider the structural and textual features but also their dependencies.

Initially, IPTR regards the text appearing in each extracted path in structural features as natural language text. The words in the text are tokenized and stemmed accordingly. Our tokenization process also considers camel case, spaces, and underlines to accommodate code-specific language. Further, IPTR does not remove and tokenize numeric values, such as hard-coded floating points and integers, as they might be integral to the program representation. IPTR vectorizes these generated tokens based on the Bag-of-words model (BoW). One could also resort to more sophisticated embeddings, such as word2vec (W2V) (Mikolov et al., 2013b). However, in programming languages the structural features are more important than the textual ones and word order can be ignored to accommodate different programming styles. Besides, we only compare text for each single path type, which means the amount of words is very few for each similarity calculation. Same reasons hold for more complex language model like BERT (Devlin et al., 2019). BoW is sufficient for this process. In our experiment, W2V does not show worthwhile improvements, but rather introduces extra training time for building the language model.

To avoid repeated computation for every new input and improve the efficiency, IPTR splits the process of building BoW models. The basic part of this process is generating word statistics. IPTR counts the words of each candidate program and stores the results during the offline phase. In the online phase, IPTR only needs to run word statistics on the input program and build the BoW model.

Table 3 shows the final comprehensive representation of the program example from Figure 2. It consists of three components. First is the list of path types that appear in this program. There is a look-up table in IPTR to maintain all types of paths so that each path from a program is described by its path type ID $P_i$, as shown in Table 3. It should be noted that path {ifStatement, {identifier, literal}} and path {ifStatement, {literal, identifier}} in Figure 2 are regarded as the same path type $P_2$ because the order of end nodes is not relevant. The second component of the feature representation is the frequency of different path types in a program. The third component contains the structure-dependent textual features. IPTR integrates the information of relative position of text and structure into the feature representation. As shown in Table 3, the text statistics with respect to each path type are stored in the final representation.

### A.4 EXAMPLE OF USER FEEDBACK

Figure 8 is an example of user feedback module. The input code is an Ackermann function in C++. However, IPTR retrieved a greatest common divisor function as its Python translation. The ground truth is shown in Figure 8 in blue. The possible reasons are the variable names are different in the ground truth and the weight of each feature element is not assigned properly. As soon as user corrects the first wrong line (change return n to return n+1), IPTR constructs feature representation for the corrected code. Then IPTR compares it with the feature representation of the input code and summarizes user's corrections. Based on this, IPTR update the query. In the example in Figure 8, user's correction emphasizes the path type {assignmentExpression, {identifier, literal}} and token 1 so that the weight of these two feature elements will be increased accordingly. Finally, with the updated query, IPTR will run a new round of retrieval. With higher weights on structural feature and token 1 and lower weights on variable names, the ground truth will be more likely to be retrieved, and in this example it does so.

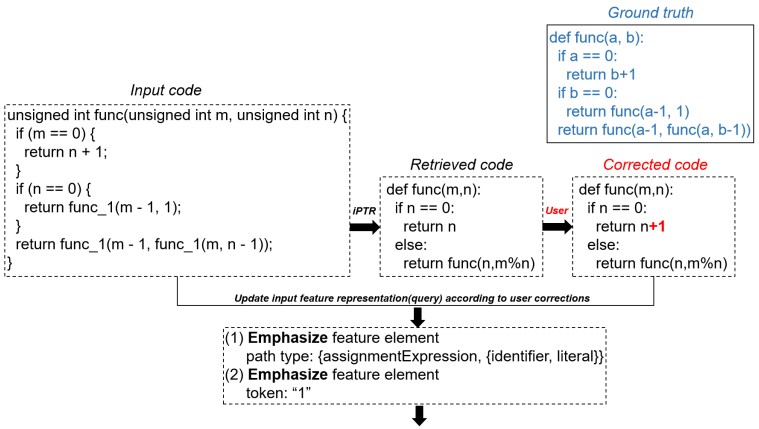

Figure 8: An example of user feedback

Table 4: Statistics of the real-world dataset

| Size | Files | Lines | Methods/Functions |
|------|-------|-------|-------------------|
| 3.8GB | 280,128 | 75,567,192 | 2,023,546 |

## A.5 EXPERIMENT ON A LARGE REAL-WORLD DATASET

**Dataset.** We choose the four programming languages with the most pushes on github - JavaScript, Python, Java, and C++. They are also representative of programming languages with different characteristics. Based on the number of stargazers, we pick 1% files in these four languages from PGA to be our raw dataset. Theoretically, IPTR works for programs of any length, but considering the practical value and the intuitiveness of the validation process, we aim to translate programs at method level in our experiment. Longer programs whose complete translations are not existing in the database can be translated by merging translations of each part. We split all the files into methods or functions as the input/output of IPTR. One limitation of PGA is that there are duplicate files across different repositories. To ensure the quality of the dataset and increase the efficiency of our program translation task, we gradually remove duplicates at file level by taking hashes of these programs and comparing their hashes. In addition, we also remove the data that cannot be successfully parsed due to format, errors, version compatibility or other issues. Since our approach does not require additional explanation except the program itself, we remove annotations and descriptions. Finally, we obtain a dataset with the size of 3.8GB as shown in Table 4.

**Experiment.** In this experiment, we only evaluate the basic program translation retrieval module PTR without user interaction to show the feasibility of the translation retrieval idea. Since there is no labeled data, we can not evaluate the translation accuracy automatically. As an alternative, we carry out sampling inspection and manually judge the correctness of the results based on the *program accuracy* stated above. Specifically, we take a random number $x \in [20, 30]$ and randomly sample $x$ programs from the dataset as a small input set. PTR translates each program in this set for one round of experiment. Then we conduct 10 rounds of experiments to obtain the average result. In the results, PTR returns the top 10 candidates with highest overall similarity for each input program.

To show the advantage of our representation method, we compare its results to other methods:

1. **PTR$_{\text{STRUCTURAL}}$:** Only the structural part of PTR.
2. **PTR$_{\text{TEXTUAL}}$:** It extracts all the text from the program and uses BoW model to only construct representation for textual features. This is the common method used in cross language code clone detection (Cheng et al., 2016; 2017). We follow the approach of Cheng et al. (2017). They implement a sliding window with a length of 20 tokens and compare the similarity of these token sequences.
3. **PTR$_{\text{S+T}}$:** This representation is the combination of (1) and (2) without the feature dependency described in Appendix A.3.
4. **PTR$_{\text{WORD2VEC}}$:** Same as PTR except changing the textual feature from Bag-of-words (BoW) to Word2vec (W2V). We train cross language word vector on PGA database.

Table 5: Comparison of different representations

(a) JS as source language

| Representation | Program accuracy | | | Mean reciprocal rank | | |
|---|---|---|---|---|---|---|
| | C++ | Python | Java | C++ | Python | Java |
| PTR$_{STRUCTURAL}$ | 53.1% | 39.8% | 46.7% | 0.74 | 0.68 | 0.73 |
| PTR$_{TEXTUAL}$ | 48.0% | 23.4% | 31.7% | 0.36 | 0.42 | 0.57 |
| PTR$_{S+T}$ | 56.2% | 43.7% | 49.9% | 0.80 | 0.71 | 0.83 |
| PTR$_{CODE2VEC}$ | 51.0% | 32.0% | 40.2% | 0.53 | 0.62 | 0.61 |
| PTR$_{WORD2VEC}$ | 61.4% | **53.9%** | **60.2%** | 0.81 | 0.83 | 0.89 |
| **PTR** | **61.4%** | 52.3% | 59.8% | **0.88** | **0.84** | **0.89** |

(b) Python as source language

| Representation | Program accuracy | | | Mean reciprocal rank | | |
|---|---|---|---|---|---|---|
| | JS | C++ | Java | JS | C++ | Java |
| PTR$_{STRUCTURAL}$ | 41.8% | 45.5% | 43.3% | 0.73 | 0.73 | 0.75 |
| PTR$_{TEXTUAL}$ | 39.5% | 31.2% | 40.9% | 0.63 | 0.58 | 0.64 |
| PTR$_{S+T}$ | 44.3% | 46.7% | 43.8% | 0.71 | 0.73 | 0.76 |
| PTR$_{CODE2VEC}$ | 41.9% | 43.4% | 43.1% | 0.66 | 0.53 | 0.71 |
| PTR$_{WORD2VEC}$ | 47.9% | **54.7%** | **48.3%** | **0.80** | **0.87** | 0.79 |
| **PTR** | **48.4%** | 54.2% | 47.9% | 0.79 | 0.85 | **0.77** |

(c) Java as source language

| Representation | Program accuracy | | | Mean reciprocal rank | | |
|---|---|---|---|---|---|---|
| | JS | Python | C++ | JS | Python | C++ |
| PTR$_{STRUCTURAL}$ | 53.3% | 47.9% | 62.2% | 0.84 | 0.76 | 0.85 |
| PTR$_{TEXTUAL}$ | 45.1% | 33.9% | 50.7% | 0.71 | 0.61 | 0.55 |
| PTR$_{S+T}$ | 54.7% | 51.6% | 67.1% | 0.88 | 0.78 | 0.86 |
| PTR$_{CODE2VEC}$ | 50.0% | 38.6% | 63.4% | 0.82 | 0.77 | 0.83 |
| PTR$_{WORD2VEC}$ | 59.1% | 57.1% | 67.7% | 0.87 | **0.84** | 0.88 |
| **PTR** | **60.6%** | **57.5%** | **69.1%** | **0.91** | 0.82 | **0.92** |

(d) C++ as source language

| Representation | Program accuracy | | | Mean reciprocal rank | | |
|---|---|---|---|---|---|---|
| | JS | Python | Java | JS | Python | Java |
| PTR$_{STRUCTURAL}$ | 59.6% | 55.8% | 63.9% | 0.78 | 0.76 | 0.82 |
| PTR$_{TEXTUAL}$ | 54.4% | 43.1% | 49.4% | 0.65 | 0.41 | 0.59 |
| PTR$_{S+T}$ | 59.7% | 57.4% | 64.8% | 0.83 | 0.79 | 0.86 |
| PTR$_{CODE2VEC}$ | 58.9% | 47.5% | 64.4% | 0.71 | 0.53 | 0.88 |
| PTR$_{WORD2VEC}$ | 64.0% | **63.6%** | 66.9% | 0.89 | 0.87 | 0.91 |
| **PTR** | **64.0%** | 63.5% | **67.0%** | **0.90** | **0.87** | **0.93** |

5. **PTR$_{CODE2VEC}$**: Code2vec is the state-of-the-art code representation which transforms code syntax tree to a vector trained by neural network. For Java, we directly use the provided trained model to generate code vectors (web, a). For other languages, we train code vectors on PGA database. Then we determine the candidates by calculating the cosine similarity.

In this experiment, we evaluate the top pick and the top 10 most likely candidates, for which we calculate the mean reciprocal rank (MRR).

**Results.** In Tables 5a-5d, we observe that our approach can successfully translate 58.8% programs between four popular languages. Note that this is the amount of successfully translated programs. Considering PTR is positioned as a translation support system, the accuracy is considerable. Translation tasks that cannot be fully automated can still be assisted by the results returned by PTR. Moreover, When the translations do not appear at top 1 in the retrieved results, they are very likely to appear at a higher rank in the candidates list. The mean reciprocal rank is around 0.8 and 0.9.

In general, compared to the other program representations, our novel representation performs significantly better. Unique structural features of program languages play a more important role than textual features in programs. And the dependency of structural features and textual features can effectively enrich the representation. W2V does not contribute much to the results and introduces extra model training time compared to simple BoW model. But we also observed W2V can help in some cases where the text of the original code is quite different from its translation. In IPTR we retain BoW for trade-off. Code2vec also considers the structure of programming languages. However,

their model can only be trained within same language, which makes it perform less satisfactory on cross-language similarity comparison.

We can also infer from the results that it is generally easier to find translation candidates for grammatically similar programming languages, such as C++ and Java. Finding translations for C++ as a low-level basic programming language can lead to higher accuracy than finding translations for Python as a high-level dynamic programming language.

