# OpenReview forum: "iPTR: Learning a representation for interactive program translation retrieval"
_ICLR.cc/2021/Conference — Reject_

### Official Review · AnonReviewer2 · 2020-10-27
**Good paper**

**Rating:** 6
**Confidence:** 3

**Review:**

This paper proposes a program translation (retrieval) method by combining  syntax tree, transformer with specific encoder/decoder, and interactive signals from user. The novelty of each used techniques is limited, but they are reasonable for the code translation task with high accuracy, even for unsupervised version without interactive signals.

The experimental results are solid and demonstrate the power of representation of the proposed variant of syntax tree. The idea of AE/AD with specific language is quite simple, but it seems that the QTM successfully transfer the representation cross different languages. The performance gain achieved by interactive signals is reasonable.

Questions:
- there are other structural features of programming language, such as graph, which can provide rich information of the code snippets rather than syntax tree.
- Is the proposed method a better code representation? Could it be applied to other downstream tasks after pre-training?
- Will the authors release their code to public?

---

> ### Author Response · Authors · 2020-11-16
> **Remarks on Novelty and Answers to Questions of the Reviewer**
>
> We thank you for your positive remarks and constructive feedback . We hope that our reply motivates you to support the acceptance of our paper.
>
> Please find our response below:
>
> **1. “The novelty of each used techniques is limited, but they are reasonable for the code translation task with high accuracy, even for unsupervised version without interactive signals.”**
>
> **Our reply:**
> We would like to point out why we believe that the novelty of the proposed techniques is high, i.e., extending state-of-the-art. The novelty is in the composition of our representation transformation pipeline that did not exist like this before and is not obvious.
> - Using autoencoding decoding directly does not work
> - Using only retrieval through similarity does not work
> - Using only the general representation for retrieval only marginally works
> - Using autoencoder/decode on the generalized representation seals the deal.
>
> While in hindsight the approach might appear straight forward, to come up with the composition was not obvious. Please consider that so far research resorts to heavily supervised techniques.
>
> **2. "there are other structural features of programming language, such as graph, which can provide rich information of the code snippets rather than syntax tree."**
>
> **Our reply:**
> Generating control flow graphs(CFG) from syntax trees is a promising idea. However as it adds more complexity, we refrained from it for the sake of efficiency. One can also extract paths from CFG to reduce the complexity. In fact, the control flow information is already included in the syntax tree. A CFG is usually extracted from an abstract syntax tree. In addition, the syntax tree contains other more information such as syntactic structure, semantics, text information, etc. Thus, we pick the most informative features - syntax tree as the basis of our program representation.
>
> **3. "Is the proposed method a better code representation? Could it be applied to other downstream tasks after pre-training?"**
>
> **Our reply:**
> Since we propose a code representation that is general to multiple kinds of programming language, there are several areas it could be used for. Due to its succinct form that we designed for the code retrieval scenario, it is also suitable for components in code database management, such as code indexing. Further applications that benefit from this representation are code plagiarism detection, code writing suggestion/auto-completion, program analysis, code task classification, etc.
>
> **4. "Will the authors release their code to public?"**
>
> **Our reply:**
> Yes we intend to publish the code right after the acceptance. We can provide you with a temporary link if desired right now.
>
> Please let us know whether we should embed any of the above responses to our paper. In the final version will for sure pick the discussion on control flow graph and other applications.
>
> Please note that our approach can be applied on different granularities of code so that smaller parts which are more likely to be shared across programs can be discovered. In our experiments, the average number of lines of each input code fragment is around 37.

---

> > ### Comment · AnonReviewer2 · 2020-11-23
> > **Reply**
> >
> > 1. Novelty
> >
> > I'm still not entirely convinced of the novelty claim that you argued for.  While it's true that you have done extensive experimical study on the representation transformation (which is obviously one of the contribution of this paper), the proposed method is relatively straight forward and has pretty limited scope.
> >
> > 2. Structural feature
> >
> > I do ackledgement what you say, but I can't fully agree your statement by ''In fact, the control flow information is already included in the syntax tree. A CFG is usually extracted from an abstract syntax tree''. In other word, can I say "the information of AST is already included in source code (,therefore AST is uncessary for code representation)"?
> >
> > I think a detailed investigation over the motivation is needed.

---

> > > ### Author Response · Authors · 2020-11-24
> > > **Reply to the questions (Update Section 3.1 in the revised version)**
> > >
> > > Thank you very much for your further suggestions.
> > > - About the novelty, our program representation can be implemented for many other applications. But of course we will conduct more experiments to validate its application scope in our future work. In this paper we aim to support program translation.
> > > - About the structural features, we have added a detailed discussion on why we do not pick CFGs into Section 3.1 of the paper. Please refer to the revised version. We hope that will give you a clearer picture of our motivation.

---

### Official Review · AnonReviewer1 · 2020-10-28
**Interesting work on program translation; some important details are unclear**

**Rating:** 5
**Confidence:** 4

**Review:**

This work proposes a retrieval-based approach for program translation. Existing ML/DL models for program translation typically design a decoder to directly generate the code in the target language. On the contrary, this work designs iPTR, which first computes a feature representation of the target code, then retrieves the most similar code in a large code corpus. Specifically, iPTR includes a query transformation model (QTM), which generates the feature representation of the target code, given the feature representation of the source code as the input. The feature representation includes the information of tokens in the code, and the paths in the syntax trees. The idea of using the paths is similar to the code2vec paper. QTM could be trained without parallel data between source and target codes. Specifically, encoders and decoders of different programming languages could be trained in a similar way to the training of autoencoders. This idea is similar to TransCoder. Meanwhile, they show that iPTR could be trained with active learning and user feedback, where they use active learning to acquire limited parallel training data, and user feedbacks are corrections of the wrong output code. They compare their approach to existing ML/DL program translation models, as well as code retrieval systems. They show that iPTR performs better than other baselines, even without active learning and feedbacks. Not surprisingly, active learning and incorporating user feedbacks further improve the performance.

Program translation is an important application, and the authors did a good job of evaluating on existing benchmarks and comparing with different types of baseline models. Intuitively, it makes sense that retrieval could at least provide more syntactically and semantically correct code, while synthesis models may struggle with generating coherent code. However, I have a couple of questions about the assumption of the task setup and the implementation of the algorithm, and I list them below.

1. It seems that the feature representation is summarized per complete code snippet for translation. Therefore, do the authors assume that the target code always exists in the retrieval corpus as a single piece of code? If this is the case, one clear limitation is that the retrieval approach could only search for existing code in the corpus, while the synthesis model could combine lines from different code snippets to construct the output code.

2. How do you simulate users for active learning and user feedback? For active learning, do you ask people to annotate the ground truth output programs for input programs, and then add them into the corpus for retrieval? For user feedback, in some parts of the paper you mention that the user modifies the extracted features of the code, but sometimes you also say that the user corrects the first wrong line. Could you provide some concrete examples of how you incorporate human annotations in training and inference loops?

3. More concrete examples of how to sample code for active learning could help. Now the description in Section 3.2.2 is not clear. For example, why N_s is 5 instead of 4? What is the benefit of "selecting program data where the largest disagreement occurs among those sampling strategies"? Do you compare among the entire training corpus?

4. What are the numbers of different paths and tokens for feature representation of code? Are text tokens representing terminal nodes or non-terminal codes? For example, are variable names included in tokens, or are they simply denoted as "identifier"? My understanding is that the feature vectors follow the bag-of-words representation, thus I don't quite understand why this simplified representation could work better than code2vec or other more advanced program encoders. The authors say that "theWord2vec and Code2vec variants of PTR cannot outperform PTR with our proposed feature representation based on structural features", but I think code2vec encodes very similar structural features in a potentially more concise way.

Writing comments: the paper requires proofreading, and there are a couple of typos. For example, (1) "to approximate measure the informativeness of the given program" on page 4: "approximate" should be "approximately". (2) "To reflect this in out feature representaion" on page 4, "out" should be "our".

---

> ### Author Response · Authors · 2020-11-16
> **[2/2]Clarifying details and answering to questions(Update Section 3.1&3.2.2 and Appendix A.3&A.4 to show more details and examples in the revised version)**
>
> *(following the first comment)*
>
> **4-1. “More concrete examples of how to sample code for active learning could help. Now the description in Section 3.2.2 is not clear. For example, why N_s is 5 instead of 4?”**
>
> **Our reply:**
> ***We revised Section 3.2.2 to better explain our sampling strategies***. We have to apologize for a writing mistake. N_s is 4.
>
> **4-2. “What is the benefit of "selecting program data where the largest disagreement occurs among those sampling strategies"? Do you compare among the entire training corpus?”**
>
> **Our reply:**
> Selecting examples where the largest disagreement occurs among those sampling strategies is the intuition for the query by committees method. General uncertainty sampling tends to be biased towards the actual learner and it may miss important examples which are not in the sight of the estimator. This is fixed by keeping several hypotheses at the same time and selecting examples where disagreement occurs. Also as explained in Dagan & Engelson 1995: as examples are selected for training, they restrict the set of consistent concepts, i.e, the set of concepts that label all the training examples correctly. Query by committees method can learn binary concepts in cases where there exists a prior probability distribution measure over the concept class. In our experiment, we investigated the entire training corpus. But in real world case, the sampling process can be conducted dynamically. Each time a new input comes, the system can run the sampling strategies to active label the data.
>
> **5-1. “What are the numbers of different paths and tokens for feature representation of code?”**
>
> **Our reply:**
> The number of different path types is about 5,000 on average for each language pair. The number of tokens used for bag-of-words can be restricted by a hyperparameter “max_vocab” to trade off effectiveness and efficiency. Our default setting is 10,000. ***We updated the paper with these numbers in Section 3.1***.
>
> **5-2. “ Are text tokens representing terminal nodes or non-terminal codes?”**
>
> **Our reply:**
> The terminal nodes represent text tokens, i.e., text in the raw code. So all text tokens are terminal nodes. Variable names are included.
>
> **5-3. “...why this simplified representation could work better than code2vec or other more advanced program encoders. ... I think code2vec encodes very similar structural features in a potentially more concise way.”**
>
> **Our reply:**
> Our feature vectors consist of two parts: structural features and textual features.
> Why do we use bag-of-words(BoW) instead of embeddings: The main reason is in programming languages the structural features are more important than the textual ones. BoW representation is only a minor part of our feature vectors. The major part is the structural features captured through the program paths that we extract from the adapted syntax trees. And word order can be ignored to accommodate different programming styles. The number reported for PTR_word2vec in Table 1 and Table 5 confirms this finding. Besides, since we also consider the dependency of structural features and textual features, we only keep the text appearing in each extracted path to generate text features and compare text for each single path type, which means the amount of words is very few for each similarity calculation. ***We updated the description in Section 3.1 and showed more details in Appendix A.3***.
> Why does our method work better than code2vec: As explained before our method generalizes across different languages. Code2vec cannot be used on top of our general syntax tree and has to be trained per programming language. With this we lose information about matching code fragments across languages. ***We discussed this in the evaluation accordingly(Section 5.1)***.
>
> **6. “Writing comments…”**
>
> **Our reply:**
> Thank you for your careful reading. We fixed the mentioned issues and would further proofread for the next version.

---

> ### Author Response · Authors · 2020-11-16
> **[1/2]Clarifying details and answering to questions(Update Section 3.1&3.2.2 and Appendix A.3&A.4 to show more details and examples in the revised version)**
>
> We thank you for the insightful comments and hope to raise your scores with our answers and revision suggestions.
>
> Please find our response below:
>
> **1. “The idea of using the paths is similar to the code2vec. QTM could be trained without parallel data...This idea is similar to TransCoder.”**
>
> **Our reply:**
> Differences to code2vec: It is true that both iptr and code2vec use paths from syntax trees. The difference is, they use paths directly from abstract syntax trees, and train a very abstract feature vector, which is suitable for their use case: i.e. predicting program properties, such as names and expression tasks. But our target is to effectively and efficiently retrieve the potential program translation in a different language. So we need a more general representation, which covers structural and textual characteristics and is still easily indexable for efficient retrieval and incrementally adaptable for user feedback. For these considerations, we construct a variant of the syntax tree that takes the low-level CST as a basis, and take the philosophy of ASTs as an inspiration. To build an efficient retrieval system and interact with users, we construct our representation with explicit statistics on feature elements, instead of training abstract distribution vectors like code2vec.
> Differences to transcoder: Our concept is  also very different from transcoder. Transcoder trains programs from all languages together to obtain a cross-language feature vector similar to word2vec. And with this vector, it trains a translation model through denoising auto-encoding. The idea is similar to transfer learning - pretraining the model on a different task first. Then transcoder optimizes the model through back-translation. Our method does not train a translation model, we try to retrieve the potential translation in a big code database. Thus our query transformation model (QTM) only optimizes the query. Unlike transcoder, we do not train a cross-language feature vector. We train each autoencoder independently for each programming language. The input of Transcoder is the raw code, while the input of our encoder/decoder system is our program feature representation, which is an abstraction of multiple languages. ***Note that the abstraction is not sufficient on its own because depending on the source language the abstraction might slightly differ as explained in the beginning of Section 3.2***.
>
> **2. “...do the authors assume that the target code always exists in the retrieval corpus as a single piece of code?...the synthesis model could combine lines from different code snippets to construct the output code.”**
>
> **Our reply:**
> In this work, we are targeting to reuse the Big Code, so we assume there is very likely a potential translation in the database. It is true that we cannot guarantee that every program can be mapped to a complete translation inside the database, but our method can retrieve translations for code fragments of any length, and aggregate the results to provide maximum support for program translation. In our experiments, we probe the database with code of average length of 37 lines. Further, we suggest to mitigate the limitation of missing code fragments through user interaction. Combining the two paradigms of retrieval and synthesis is a promising direction but would not fit within the scope of a single paper because of the diverse set of components that need to be explained, such as the retrieval engine, query model ( in our case the QTM), the user interaction and the synthesis.
>
> **3. “How do you simulate users for active learning and user feedback? For active learning, do you ask people to annotate the ground truth...and then add them into the corpus for retrieval? For user feedback, ...Could you provide some concrete examples...?”**
>
> **Our reply:**
> For active learning, the system leverages ground truth output. But the obtained label will only be used for training the query transformation model(QTM), and will not be added into the test corpus for retrieval.
> For user feedback, simulating user feedback is not trivial. Our setup checks the results after each corrected line. To have a systematic comparison we report the overall result after the first wrong line has been corrected in a retrieved program. Note that the process of modifying the extracted features of the code will be done automatically by the system according to user’s corrections. Due to our structured feature representation, iPTR can easily transform user’s corrections on the raw code into the form of our feature representation. That’s one advantage of our representation. User does not need to learn how to construct features, he/she only needs to deal with the high-level raw code. We showed an example of user feedback in Appendix A.4.
>
> *(followed by the second comment)*

---

### Official Review · AnonReviewer4 · 2020-10-29
**Lacking significant algorithm/theory contribution**

**Rating:** 4
**Confidence:** 4

**Review:**

Summary:
This paper seeks to solve program translation problem through code retrieval. It proposes an interactive code retrieval system, called iPTR to perform cross-language retrieval with minimum code pairs. The method extracts textual features and structural features from code and transform the features into target-language features through an autoencoder, which is trained through a uni-language manner. It further leverages users' correction to update feature representation and for new round of retrieval.

The problem is challenging, especially since the parallel data is scarse. I think the idea of utilizing both textual features and structural features is interesting. The idea of training autoencoder for each language and fuse encoder and decoder of different languages is also very interesting, though questionable as mentioned below.

Concerns:
- The major concern is that this paper does not made significant novel and technical algorithm/theory contribution, but rather like proposing a system. Therefore in my opinion the paper does not fit ICLR very well.

- Due to lack of parallel training corpus for query transformation model (QTM), the paper utilizes simple uni-language autoencoder training for each language, and take the encoder and decoder from the corresponding language respectively as the transformer model. This could cause significant issue if not taking seriously. The hidden space of different language can vary drastically, and hence the encoder output is very likely to not make any sense to decoder to perform the transformation. There should be a mechanism that encourages the hidden space of different languages to the similar.

-The paper tries to solve the program translation problem with retrieval method. In the experiment, what is the database to retrieve from? Is it that all training code form the database, or is it all training and testing code form the database? In practical situation, the desired translation is very likely to not exactly match the some code in the database. Therefore, I don't understand why the program accuracy of retrieval method can achieve so high if the database doesn't overlap with ground truth codes a lot.

---

> ### Author Response · Authors · 2020-11-16
> **Clarifying misunderstanding and answering to questions (Update Appendix A.5 to show another experiment  in the revised version)**
>
> We thank you for the insightful comments and hope to raise your scores with our answers and revision suggestions.
>
> Please find our response below:
>
> **1. “The major concern is that this paper does not made significant novel and technical algorithm/theory contribution, but rather like proposing a system. Therefore in my opinion the paper does not fit ICLR very well.”**
>
> **Our reply:**
> We would like to point out that our solution that combines a generalized syntax tree representation with automatically trainable autoencoders is beyond state of the art in language translation. Please consider that other pieces of work on program representation have also been published at ICLR:
> [1] Xinyun Chen, Chang Liu, and Dawn Song. Tree-to-tree neural networks for program translation.  ICLR 2018.
> [2] Miltiadis Allamanis, Marc Brockschmidt, and Mahmoud Khademi. Learning to represent programs with graphs. ICLR 2018.
>
> **2. “Due to lack of parallel training corpus for query transformation model (QTM), the paper utilizes simple uni-language autoencoder training for each language, and take the encoder and decoder from the corresponding language respectively as the transformer model. This could cause significant issue if not taking seriously. The hidden space of different language can vary drastically, and hence the encoder output is very likely to not make any sense to decoder to perform the transformation. There should be a mechanism that encourages the hidden space of different languages to the similar.”**
>
> **Our reply:**
> There is a misunderstanding here. We did design a mechanism that makes the hidden space of different languages to be similar. It is true that directly encoding the original code from different languages will cause the hidden space to vary drastically. Therefore, we first construct a feature representation for each piece of code based on the extracted Abstract Syntax Trees. This representation is general to multiple kinds of programming languages. Then we input this general representation to the encoder instead of the original code. This method circumvents the danger you mentioned. ***The approach of constructing representation is explained in Section 3.1.***
>
> **3-1. “The paper tries to solve the program translation problem with retrieval method. In the experiment, what is the database to retrieve from?”**
>
> **Our reply:**
> We retrieve the results from the two parallel datasets with ground truth: Java-C# dataset and GeekforGeeks dataset.
>
> **3-2. “Is it that all training code form the database, or is it all training and testing code form the database?”**
>
> **Our reply:**
> We train the autoencoder for each language on a large dataset without ground truth: Public Git Archive(PGA). Because the autoencoder is an unsupervised model, there is no testing data.
>
> **3-3. “In practical situation, the desired translation is very likely to not exactly match the some code in the database. Therefore, I don't understand why the program accuracy of retrieval method can achieve so high if the database doesn't overlap with ground truth codes a lot.”**
>
> **Our reply:**
> In summary, there are two types of database. First type is a large database PGA without ground truth. Let us denote it by D1. Second type is the parallel dataset with ground truth. We have two such datasets in our experiments:  Java-C# dataset and GeekforGeeks dataset. Let us denote this type of dataset by D2. In the pre-training phase, we use D1 to train the autoencoders for QTM. We do not use D2 for any training. In the experimental phase, each time we randomly pick one piece of code from D2, then use it to retrieve its potential translation also from D2. That’s why we can get relatively high accuracy. In this work, we are targeting to reuse the Big Code, so we assume there is very likely a potential translation in the database. Our focus is how to use a piece of raw code as a query to retrieve its potential translation in target language. ***The details of our dataset are shown in the subsection “Dataset” in Section 5***.
> Nevertheless, we still conducted another experiment to further evaluate the effectiveness of our method: this time in the experimental phase, we randomly pick one piece of code from D1, and use it to retrieve its potential translation also in D1, where we do not know whether there is a potential translation. ***We show this experiment in Appendix A.5***. The accuracy of translating programs between four popular languages(Java, C++, Python, Javascript) is 58.8% on average. Considering our system is positioned as a translation support system, the accuracy is comparable to the accuracy of other translation techniques on easier datasets.

---

### Decision · Program_Chairs · 2021-01-07
**Final Decision**

**Decision:**

Reject

**Comment:**

I found the setup for this paper a bit contrived. The tool is presented as a code translation tool, but it really functions more as a multi-language code search tool. The Idea is that one has a program in language A, and a database that contains the same program in language B, so one can translate from A to B simply by searching for the right program in the database.

When evaluated as a language translation tool, it appears to outperform existing language translation schemes, but this is an unfair comparison, because iPTR is being given a database that contains the exact translation of the program in question. The performance is also compared with code search tools, but these are also apples-to-oranges comparisons, because the tools in question are operating from very high-level queries. A much more comparable baseline would be the Yogo tool  recently published in PLDI (https://dl.acm.org/doi/abs/10.1145/3385412.3386001), or for compiled languages you could compare against statistical similarity tools for binaries (https://dl.acm.org/doi/10.1145/2980983.2908126).

The experiment in the appendix A5 is more fair to standard language translation, and it yields results that are much less impressive. I would be much more comfortable with this paper if it were written around this experiment, or alternatively if it were evaluated against a more comparable approach for semantic code search.